# Assessing the Interactive Effects of High Salinity and Stocking Density on the Growth and Stress Physiology of the Pacific White Shrimp *Litopenaeus vannamei*

**Fei Liu [1], Jinfeng Sun [1], Jinnan Long [1], Lichao Sun [1], Chang Liu [1], Xiaofan Wang [1], Long Zhang [1], Pengyuan Hao [1], Zhongkai Wang [1], Yanting Cui [1], Renjie Wang [1] and Yuquan Li [1,2,*]**

1 School of Marine Science and Engineering, Qingdao Agricultural University, Qingdao 266109, China; liufeizn@qau.edu.cn (F.L.); 20212111012@stu.qau.edu.cn (J.S.); whrcjinnanlong@163.com (J.L.); sunlichao6699@163.com (L.S.); changliu2024@163.com (C.L.); 20222111013@stu.qau.edu.cn (X.W.); 20222211027@stu.qau.edu.cn (L.Z.); 20222111019@stu.qau.edu.cn (P.H.); zkwang@qau.edu.cn (Z.W.); yt-cui1114@163.com (Y.C.); jiezi311@126.com (R.W.)

2 Key Laboratory for Sustainable Utilization of Marine Fisheries Resources, Ministry of Agriculture, Yellow Sea Fisheries Research Institute, Chinese Academy of Fishery Sciences, Qingdao 266071, China

* Correspondence: jiangfangqian@163.com or yqli@qau.edu.cn

**Abstract:** This study was conducted to evaluate the effects of high salinity combined with stocking density on *Litopenaeus vannamei*. Three salinity gradients, namely, 28 g/L, 36 g/L, and 44 g/L, and two stocking densities, namely, 300 and 600 shrimp/m$^3$, were used to analyze the synergistic effect of high salinity and stocking density on the growth performance, digestibility, and energy budgets of *L. vannamei*. The experimental testing period lasted 45 days. The research results showed that a salinity level of 36 g/L was the most suitable salinity level for shrimp growth under both high and low stocking densities. The body weight, specific growth rate, and relative weight gain of the shrimp in the 36 g/L salinity group were significantly higher than those in the other two salinity groups under both high and low stocking densities. The high-density farming group with 600 shrimp/m$^3$ exhibited a significant inhibition of shrimp growth compared to the low-density group under the same salinity conditions. The activities of amylase, lipase, and protease in the high-density-group shrimp gradually decreased with an increase in salinity, and the three digestive enzymes had the same overall effect of changing trends. This indicates that under high-density farming conditions, the increase in salinity is not conducive to the digestive function of shrimps. At the same time, the proportion of respiratory energy to feeding energy gradually decreased in the high-density group and with the increase in salinity. However, under the same salinity conditions, the higher the stocking density, the higher the energy consumed by respiration compared to the low-density group. In addition, the expression of the growth-related gene's small nuclear ribonucleoprotein polypeptide G (*SNRPG*) under high stocking density was significantly lower than that in the low-density group at a salinity of 28 g/L, and ribosomal protein L7 (*RPL7*) expression was also significantly lower under high stocking density than that in the low-density group at a salinity of 44 g/L. The expression levels of molting-related genes retinoid X receptor (*RXR*), ecdysone receptor (*ECR*), and ecdysone-induced protein 75 (*E75*) were significantly higher in the 36 g/L salinity group compared with the other two salinity groups under high-stocking-density treatment. The findings indicate that the synergistic effects of salinity and stocking density have a significant impact on the growth of *L. vannamei*, and excessive salinity would inhibit its growth in the process of high-density culturing.

**Keywords:** high salinity; stocking density; growth; energy budget; *Litopenaeus vannamei*

**Key Contribution:** Salinity and stocking density have significant synergistic impacts on the Pacific white shrimp. Excessive salinity and stocking density have negative effects on shrimp aquaculture.

## 1. Introduction

As a widely farmed aquatic economic species, the Pacific white shrimp (*Litopenaeus vannamei*) has been popular with consumers since it was introduced to China in 1988 [1]. It has a wide range of salinity characteristics, living in freshwater, brackish water, saltwater, and other water bodies, and it can be used for factory farming [2]. In intensive aquaculture, the determination of the maximum breeding capacity of a water body is particularly important to maximize the benefits per unit of water due to the restricted water area. High rearing density increases competition for food and living space between individuals and increases individual growth variability, and among crustaceans, it is generally accepted that culture density is negatively correlated with shrimp growth rate and weight gain rate [3]. Shrimp growth is affected by culture density in two main ways. On the one hand, the process of increasing culture density leads to lower dissolved oxygen levels and higher accumulation of toxic substances such as inorganic phosphorus and ammonia and nitrogen in the water environment, and these physicochemical factors generate stress on shrimp [4,5]. For example, it has been found that culture density significantly affects bioflocs and various environmental factors, and related studies have shown that culture density indirectly affects aquatic organisms by influencing environmental factors such as DO, pH, TAN, $NO_2$-N, and $NO_3$-N in the water column [6–8]. On the other hand, from the behavior and physiology analysis of prawns, the increase in culture density will compress the survival space of the shrimp, and the frequent displacement of shrimp leads to frequent exhibition of fighting behaviors and an increase in the rate of interspecific mutilation [9–11]. Due to the high activity, the proportion of energy consumed through respiration increases, while the proportion of energy required for growth decreases as a percentage of the energy produced by decomposing feed, in turn leading to a decrease in growth efficiency [3,12]. High-density culturing can have an impact on the normal growth and survival of shrimp [13,14].

Environmental factors affect the adaptability of organisms to changes in environmental conditions [15–18]. Salinity is a fundamental environmental factor in mariculture, and it has a more significant influence in intensive culture patterns. Abnormal changes in salinity can affect many physiological functions of shrimp [19–22]. Lemos found that the energy efficiency of *Farfantepenaeus paulensis* reached a maximum at a salinity level of 25 g/L [23]. It has been shown that increasing salinity within a certain range enhanced the growth of the Pacific white shrimp [24]. In addition, studies have shown that shrimp grow better at a salinity level of 20 g/L than at 13 g/L [25]. Salinity stress causes changes in the digestive function of an organism by affecting osmotic pressure and thus the activity of related digestive enzymes.

The activity of pepsin, tryptase, and alkaline phosphatase in *Macrobrachium mipponensis* is best at a salinity of 14‰ when compared with salinity levels of 7‰ and 20‰ [26]. The cited experiment also showed that the mRNA expression in the hepatopancreas of *L. vannamei* also decreased under low-salt conditions [20]. In addition, the activity of trypsin in shrimp decreased at higher salinity, whereas chymotrypsin activity was reversed [27]. In addition to salinity stress, studies have confirmed that the activities of trypsin, amylase, and lipase of *Palaemonetes sinensis* were significantly reduced under high-density culturing [3]. It has been found that crustaceans use more energy for growth under suitable salinity conditions. Furthermore, salinity can affect the physiological activity of crustaceans in conjunction with environmental factors such as temperature and density [28–30].

The above studies show that salinity and culture density are significant factors in *L.vannamei* culturing. Asia has vast salt fields and high-salinity environments that can be developed for shrimp farming. However, excessive salinity and culture density may have adverse effects on shrimp. Therefore, it is necessary to explore suitable farming salinity and density. In our study, we aimed to analyze the combined effects of the growth, digestion, and energy balance of whiteleg shrimp under high salinity and culture density stress to provide basic data for intensive culture and to promote the healthy culturing of shrimp with high productivity and efficiency.

## 2. Materials and Methods

### 2.1. Experimental Animals

The experimental shrimp were obtained from a commercial farm in Haiyang Shandong Province. The average weight was $2.10 \pm 0.20$ g. The shrimp were cultured in tanks with seawater (salinity 23 g/L, pH $7.5 \pm 0.5$) and kept at $28 \pm 0.5$ °C for seven days before the formal experiment. The tanks used were cylindrical PVC tanks with a radius of 90 cm and a height of 100 cm. Feeding was conducted three times a day (8:00, 14:00, and 22:00). Half the volume of the aquaculture water body was changed every day, and the dissolved oxygen level was not less than 6 mg/L. Qingdao Agriculture University's Animal Experiment Ethics Committee approved all treatments in this study.

### 2.2. Experimental Design and Samples Collection

After acclimation, the shrimp were randomly divided into 6 groups, and each group contained three replicates. The stocking densities were set at 300 shrimp/m$^3$ and 600 shrimp/m$^3$, and each stocking density had three salinity levels: 28 g/L, 36 g/L, and 44 g/L. Among them, the one with a stocking density of 300 shrimp/m$^3$ and a salinity of 28 g/L was set up as a control group. The salinity of these experimental groups was increased by 3–4 g/L/day to reach 36 g/L and 44 g/L, respectively. The salinity conditions required for the experiments were induced via sea salt addition. The YSI 556MPS Handheld Multiparameter Meter (YSI, Inc., Yellow Springs, OH, USA) was used for seawater parameter detection. The experiment lasted 45 days. Half the volume of farmed water was changed every day. The shrimp were fed twice a day (at 8:00 and 20:00), regularly providing an excess of feed. The excess feed left by the shrimp was collected after 1.5 h of feeding, and feces and shrimp shells were collected after 4 h of feeding. The collected samples were dried at 70 °C and stored for nitrogen and energy balance measurement. The culture experiment lasted for 45 days, and the body length and weight of 6 shrimp in each box were randomly measured weekly. Hepatopancreas, muscle, and stomach were collected at 0 d and 45 d, and a total of nine shrimp were sampled per treatment and per time point (three replicates, with three shrimp per replicate). Tissues were frozen in liquid nitrogen rapidly and stored at $-80$ °C.

### 2.3. Determination of Enzyme Activity

The hepatopancreas and muscle were made into 10% tissue homogenate in PBS. The homogenate was centrifuged at 4 °C at 12,000 rpm, and the supernatant was collected and used for the determination of enzyme activity. The activity levels of enzymes such as pepsin, amylase, and lipase were detected using commercial assay kits (Nanjing Jiancheng Institute, Nanjing, China). The experimental method was executed according to the instructions in the kits' manuals.

### 2.4. Quantitative Real-Time PCR

In order to investigate the effects of high salinity and stocking density stress on genes expression related to growth, molting, and energy budget, three growth-related genes, namely, *SNRPG*, *L18*, and *RPL7*; three molting-related genes, i.e., *RXR*, *ECR*, and *E75*; and an energy-related gene, *ATP-α* subunit, were selected for quantitative real-time PCR (qPCR) analysis. The *β-actin* gene of *L. vannamei* was selected as an internal standard. Information on the primers used for qPCR is shown in Table 1.

The total RNA of the samples was extracted using TRIzol Reagent (Vazyme, Nanjing, China). The cDNA was synthesized using a PrimeScript™ RT Reagent Kit (Vazyme, Nanjing, China). The qPCR reaction was carried out according to the manufacturer's instructions included with the ChamQ™ Universal SYBR® qPCR Master Mix Kit (Vazyme, Nanjing, China). The qPCR reaction was performed using a CFX96™ Real-Time System (BIO-Rad, Hercules, CA, USA). The $2^{-\Delta\Delta CT}$ comparative CT method was used for data analysis [31].

**Table 1.** Information on primers used in this study.

| Genes | NCBI Accession Number | Forward (5′-3′) | Reverse (5′-3′) |
|---|---|---|---|
| *L18* | XM_027372553.1 | GAAGTGCCCAAGATGACCG | GACCCTGGATGAGGACTGTGT |
| *SNRPG* | XM_037920053.1 | GTGGATGATGGGGTGGAAGT | CATTTGGGTTTGGGACTACGA |
| *RPL7* | XM_027357991.1 | CTTCATCACCTGGGGTTATCC | GAACACGGCGTCCATCAAT |
| *RXR* | XM_027379485.1 | ACCACCCTACAATGATGACGAA | GGATGGCTCGCTTGACTCTC |
| *EcR* | XM_027356276.1 | CTGACGACGACTCTGAAGATCC | TGCCTTGAGGAGTGTAATCTGG |
| *E75* | XM_027355294.1 | CCATGCAACCCACCGTAAC | GAGCACCCAAGCCTGAATGT |
| *ATPase-α* | XM_027356918.1 | CTCTTGTGCCTATTGGTCG | CGTTGAATCGCTTCTGGT |
| *β-actin* | XM_062036902.1 | GCCCTGTTCCAGCCCTCATT | ACGGATGTCCACGTCGCACT |

In the above table: *L18*, ribosomal protein L18; *SNRPG*, small nuclear ribonucleoprotein polypeptide G; *RPL7*, ribosomal protein L7; *RXR*, retinoid X receptor; *EcR*, ecdysone receptor; *E75*, ecdysone-induced protein 75; *ATPase-α*, ATP synthase subunit alpha; *β-actin*, beta-actin gene.

*2.5. Growth Performance*

The weight gain rate (WGR), specific growth rate (SGR), survival rate (SR), and length gain rate (LGR) were calculated with the following equations:

$$\text{WGR (\%)} = [W_t - W_0]/W_0 \times 100\%,$$

$$\text{SGR (\%/d)} = [\ln W_t - \ln W_0]/t \text{ (d)} \times 100\%,$$

$$\text{SR (\%)} = N_t/N_0 \times 100\%,$$

$$\text{LGR (\%)} = [(L_t - L_0)/L_0] \times 100\%,$$

where $W_t$ is final mean body weight (wet weight), $W_0$ is initial mean body weight (wet weight), t is the number of days in culture, $N_t$ is the final number of shrimp, $N_0$ is the initial number of shrimp, $L_t$ is final mean body length, and $L_0$ is initial mean body length [1,32].

*2.6. Nitrogen Balance*

The nitrogen content of feed, whole shrimp, shells, and feces (nitrogen content = protein content × 0.16) was determined using a Kjeldahl azotometer. The nitrogen balance equation is shown below [33,34]:

$$C_N = G_N + E_N + U_N + F_N = A_N + F_N,$$

where $C_N$ is the intake of feed nitrogen ($mg \cdot g^{-1} \cdot d^{-1}$), $G_N$ is the nitrogen accumulated in whole shrimp ($mg \cdot g^{-1} \cdot d^{-1}$), $E_N$ is molting material consumed ($mg \cdot g^{-1} \cdot d^{-1}$), $U_N$ is nitrogen lost in excretion ($mg \cdot g^{-1} \cdot d^{-1}$), $F_N$ is nitrogen excreted in feces ($mg \cdot g^{-1} \cdot d^{-1}$), and $A_N$ is absorbed nitrogen ($mg \cdot g^{-1} \cdot d^{-1}$).

*2.7. Energy Balance Measurement*

Feed, whole shrimp, feces, and molting shells were first dried at 70 °C. Then, an IKA-C200 (IKA, Staufen, Germany) was used to measure gross energy. The energy budget was calculated according to the following formula:

$$C = G + E + F + R + U,$$

where C is the energy consumed through feeding (kJ), G is the energy spent on growth (kJ), F is the energy lost in feces (kJ), U is the energy lost in excretion (kJ), E is the energy spent on exuviae (kJ), and R is the energy expenditure for respiration (kJ). The estimate of U was based on the nitrogen budget equation, A = G + R, wherein A is assimilation energy ($kJ \cdot g^{-1} \cdot d^{-1}$) [24,35].

The estimation of U was based on the nitrogen budget equation:

$$U = (C_N - G_N - F_N - E_N) \times 24.83,$$

where $C_N$ is the nitrogen consumed from feed (g), $F_N$ is the nitrogen lost in feces (g), $G_N$ is the nitrogen deposited in a shrimp's body (g), $E_N$ is the nitrogen lost through molting (g), and 24.83 is the energy content in excreted nitrogen per gram (KJ/g) [32,36].

The value of respiratory energy (R) was calculated using the following equation [37]:

$$R = C - G - F - E - U.$$

### 2.8. Statistical Analysis

All data for the experiments are expressed as means with standard deviations. A two-way ANOVA followed by Tukey's honest significant difference test ($p < 0.05$) were used to analyze significant differences between the observations of the control and test groups. The statistical analyses were conducted using SPSS 18.

## 3. Results

### 3.1. Digestive Enzyme

The activity of pepsin in the hepatopancreas significantly decreased in the high-density treatment as salinity increased (Figure 1A), while pepsin activity in the low-density treatment showed a change rule consisting of a gradual decrease followed by a gradual increase ($p < 0.05$). Among the stocking density experimental treatments, the activity of pepsin in the 28 g/L and 36 g/L salinity treatments was significantly higher in the high-density (600 shrimp/m$^3$) group than in the low-density (300 shrimp/m$^3$) group, while the opposite was true in the 44 g/L salinity treatments ($p < 0.05$). As shown in Table 2, salinity and culture density had a significant interactive effect on the activity of pepsin ($p < 0.05$).

**Table 2.** Summary of two-way analysis of variance: the effects of salinity (28 g/L, 36 g/L, and 44 g/L) and density (300 shrimp/m$^3$ and 600 shrimp/m$^3$) on growth performance, enzyme activity, and gene expression.

| Item | Tissue | *p*-Values | | |
|------|--------|:---:|:---:|:---:|
| | | **Salinity (S)** | **Density (D)** | **S × D** |
| *L18* | muscle | 0.439 | 0.138 | 0.693 |
| *RPL7* | muscle | 0.048 | 0.102 | 0.104 |
| *SNRPG* | muscle | 0.012 | 0.016 | 0.022 |
| *ECR* | muscle | 0.013 | 0.058 | 0.045 |
| *RXR* | muscle | 0.007 | 0.005 | 0.029 |
| *E75* | muscle | 0.011 | 0.000 | 0.001 |
| *ATP* | muscle | 0.046 | 0.039 | 0.046 |
| Protease | hepatopancreas | 0.019 | 0.493 | 0.043 |
| Amylase | hepatopancreas | 0.791 | 0.046 | 0.050 |
| Lipase | hepatopancreas | 0.185 | 0.019 | 0.261 |
| The body length | | 0.047 | 0.012 | 0.048 |
| The body weight | | 0.024 | 0.015 | 0.048 |
| Dry weight | | 0.014 | 0.015 | 0.027 |
| Relative weight gain | | 0.032 | 0.027 | 0.042 |
| Specific growth rate | | 0.028 | 0.019 | 0.037 |

The amylase activity significantly decreased with increasing treatment salinity in the high-density treatments, while amylase activity in the low-density treatments significantly increased ($p < 0.05$) (Figure 1B). The amylase activity in the low-density treatments was significantly higher than that in the high-density group in the 44 g/L salinity group ($p < 0.05$). In terms of amylase activity, salinity and stocking density interacted significantly ($p < 0.05$) (Table 2).

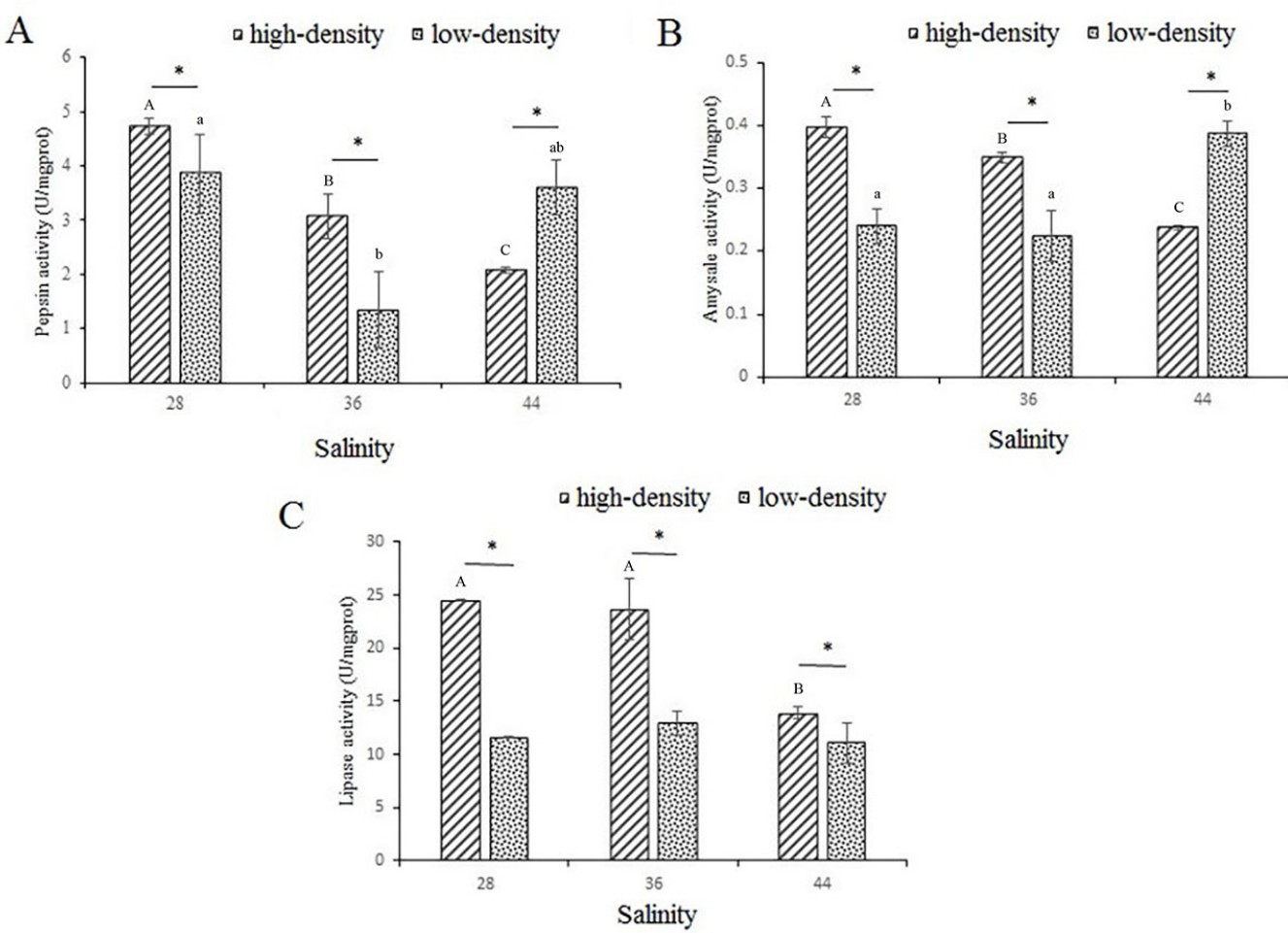

**Figure 1.** Enzyme activities related to digestion-related parameters in the hepatopancreas: (**A**) the activity of pepsin; (**B**) the activity of amylase; (**C**) the activity of lipase. Asterisks indicate significant differences among different culture densities at the same salinity ($p < 0.05$). Capital letters indicate significant differences in the different salinities under high-culture-density conditions ($p < 0.05$). Lowercase letters indicate significant differences in the different salinities under low-culture-density conditions ($p < 0.05$) (the same letters or no letter indicate that there are no significant differences, with $p > 0.05$). The asterisk (*) indicates significant differences between high-density and low-density treatments under the same salinity conditions ($p < 0.05$).

Among all the salinity groups, the activity of lipase was significantly higher in the high-density group than in the control group ($p < 0.05$) (Figure 1C), and it was significantly lower ($p < 0.05$) in the high-culture-density treatment under a salinity of 44 g/L.

*3.2. Growth*

The effects of salinity and stocking density stress on the growth results are shown in Figure 2. The shrimps' body weight (wet and dry) (Figure 2A,E), body length (Figure 2B), specific growth rate (Figure 2C), and relative weight gain rate (Figure 2D) showed a change rule consisting of a gradual increase followed by a gradual decrease as salinity increases. Almost all the growth indicators of the 36 g/L salinity group were higher than those of the other two salinity groups. And all the growth indicators in the high-density group were significantly lower than those in the control group. Our analysis showed that salinity and stocking density interacted significantly with respect to the growth traits of *L. vannamei* (Table 2). Under both stocking density conditions, the salinity of 36 was found to be the most favorable for shrimp growth.

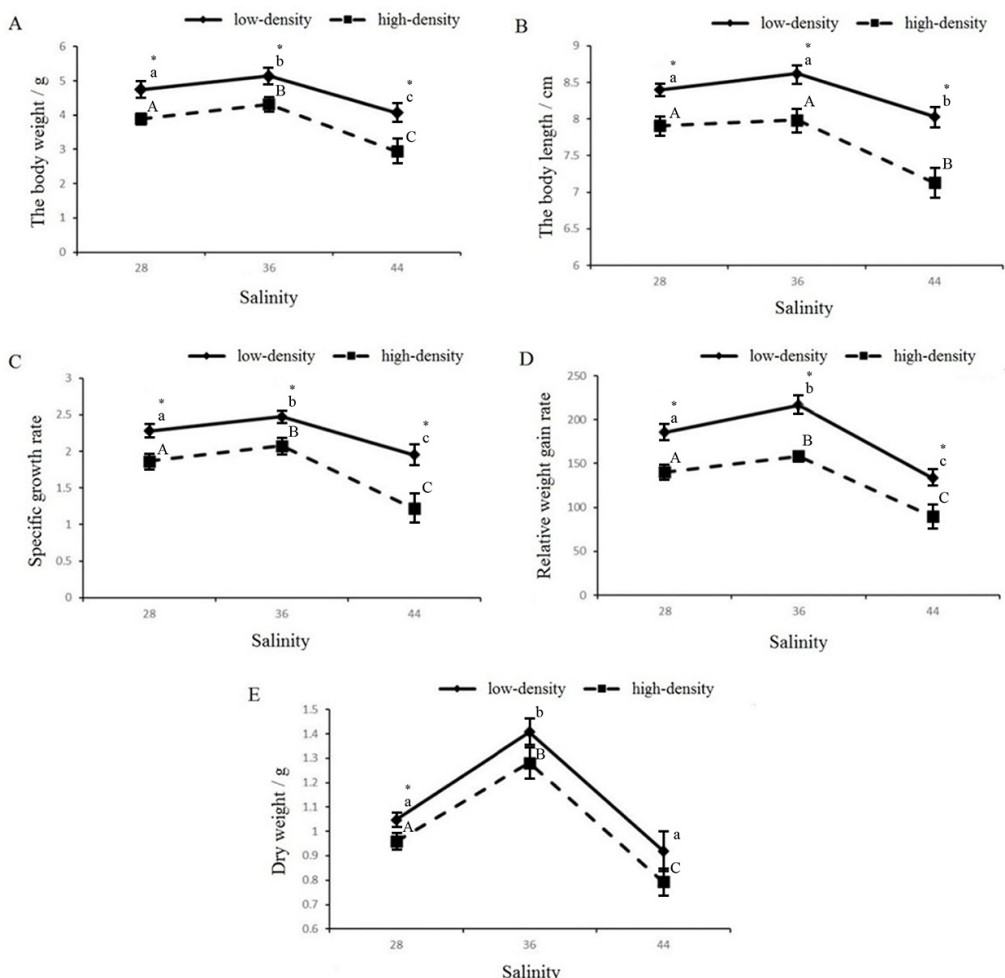

**Figure 2.** The growth-related parameters under different salinities and types of stocking density stress: (**A**) body weight (wet weight); (**B**) body length; (**C**) specific growth rate; (**D**) relative weight gain; (**E**) body weight (dry weight). Capital letters indicate significant differences in the different salinities under high-culture-density conditions ($p < 0.05$). Lowercase letters indicate significant differences in the different salinities under low-culture-density conditions ($p < 0.05$) (the same letters or no letters indicate that there are no significant differences, with $p > 0.05$). The asterisk (*) indicates significant differences between the high-density and low-density treatments under the same salinity conditions ($p < 0.05$).

### 3.3. Growth-Related Genes

The expression of *SNRPG* under conditions of high salinity and stocking density stress is shown in Figure 3A. In the low-density treatment, the expression of *SNRPG* changed significantly as salinity increased, showing a change rule consisting of decreasing and then increasing ($p < 0.05$). In the 28 g/L and 36 g/L salinity groups, the expression of *SNRPG* was significantly affected by both high and low densities ($p < 0.05$). As shown in Table 2, salinity and stocking density interacted significantly with regard to the expression of *SNRPG* ($p < 0.05$).

The expression of *L18* under conditions of high salinity and stocking density stress is shown in Figure 3B. In the salinity of 36 g/L group, the expression of *L18* reached a peak in the high-density treatments. In the salinity of 36 g/L group, high and low stocking densities had significant effects on the expression of *L18*, and the expression in the high-density group was significantly higher than that in the low-density group ($p < 0.05$).

The expression of *RPL7* under conditions of high salinity and stocking density stress is shown in Figure 3C. In the high-density treatments, there was no significant effect on the expression of *RPL7* with the change in salinity. However, in the low-density group, the expression of *RPL7* increased significantly as salinity increased, and significant differences in the expression of *RPL7* occurred between the high- and low-density groups only under the conditions of the high-salinity treatment ($p < 0.05$).

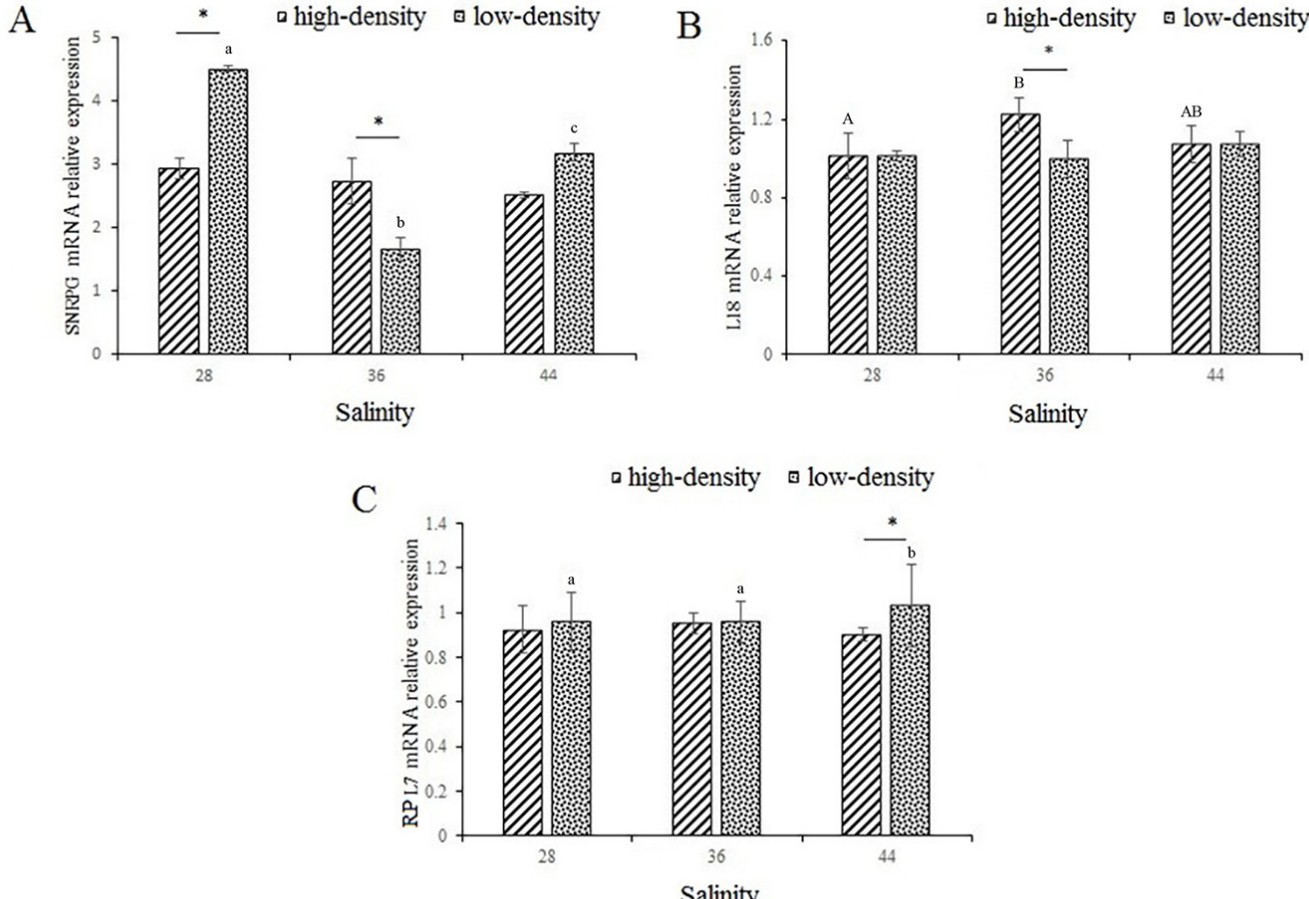

**Figure 3.** The expression of *SNRPG* (**A**), *L18* (**B**), and *RPL7* (**C**) growth-related genes in muscle. *SNRPG*, small nuclear ribonucleoprotein polypeptide G; *L18*, ribosomal protein L18; *RPL7*, ribosomal protein RPL7. Capital letters indicate significant differences in the different salinities under high-culture-density conditions ($p < 0.05$). Lowercase letters indicate significant differences in the different salinities under low-culture-density conditions ($p < 0.05$) (the same letters or no letters indicate that there are no significant differences, with $p > 0.05$). The asterisk (*) indicates significant differences between high-density and low-density treatments under the same salinity conditions ($p < 0.05$).

*3.4. Molting-Related Genes*

The expression of *RXR* under conditions of high salinity and stocking density stress is shown in Figure 4A. In the high-density treatments, the expression of *RXR* showed a significant increase followed by a decrease as salinity increased ($p < 0.05$). However, the expression of *RXR* was significantly reduced under the low-density treatment ($p < 0.05$). Among the salinity 28 g/L groups, the expression of *RXR* in the low-density group was significantly higher than that in the high-density group ($p < 0.05$).

The expression of *ECR* under high salinity and stocking density stress is shown in Figure 4B. In the high-density group, the expression of *ECR* first increased and then decreased significantly as salinity increased, but it was significantly decreased in the low-density group ($p < 0.05$). The expression of *ECR* was significantly higher under the

low-density-culture conditions than that under the high-density-culture conditions in the 28 g/L salinity group; however, the situation was reversed in the 36 g/L salinity group.

The expression of *E75* under high salinity and stocking density stress is shown in Figure 4C. Under high-density stress, the expression of *E75* increased with salinity and decreased significantly after reaching a peak at a salinity of 36 g/L ($p < 0.05$); under the low-density treatments, they decreased gradually. In the 36 g/L and 44 g/L salinity groups, the expression of *E75* was significantly higher under high-density stress than that under low-density stress ($p < 0.05$). Table 2 shows that salinity and culture density interacted significantly regarding the expression of the three genes ($p < 0.05$). The three molt-related genes tested showed increased activity in the high-density experimental group at a salinity of 36.

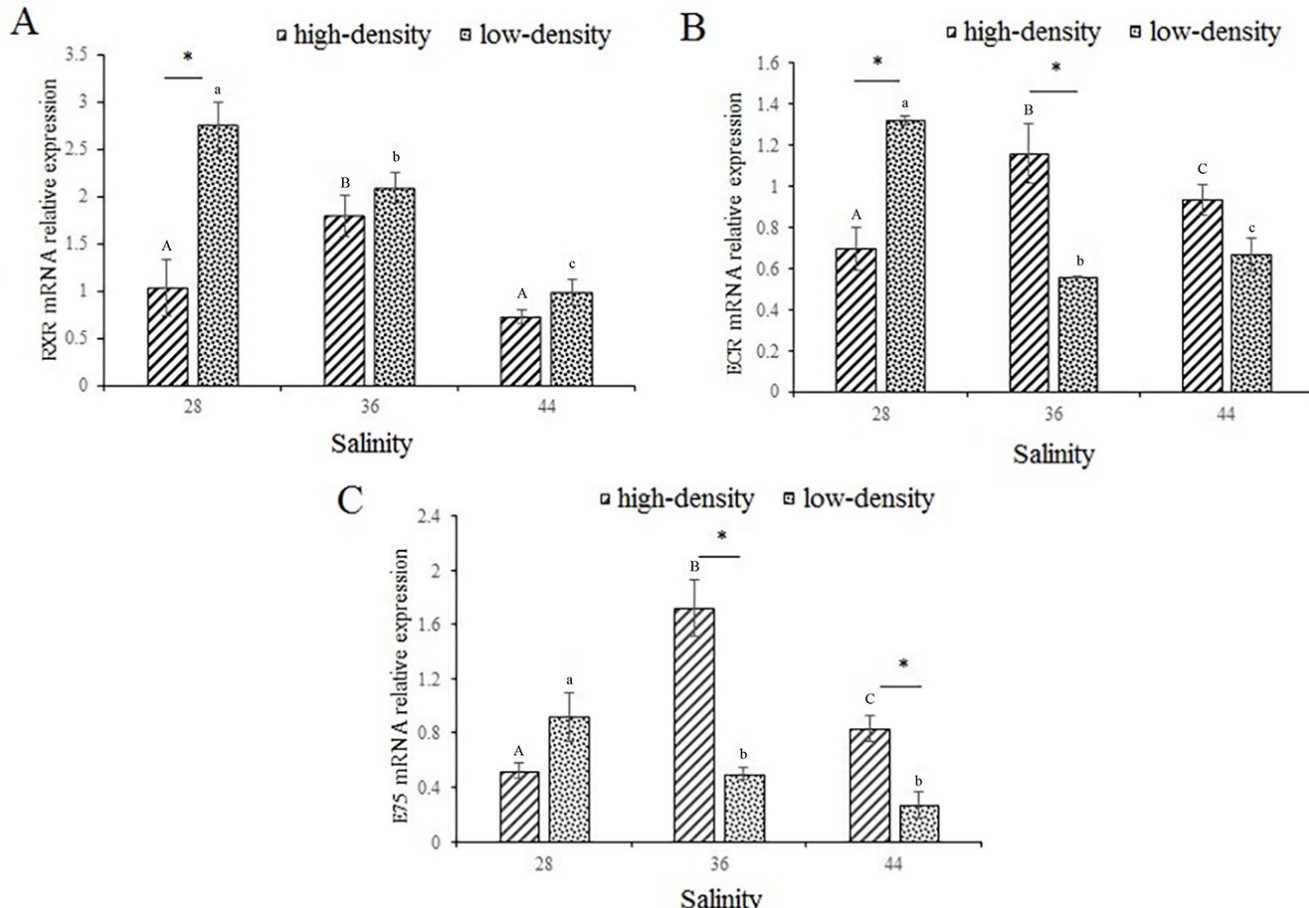

**Figure 4.** The expression of *RXR* (**A**), *ECR* (**B**), and *E75* (**C**) molting-related genes in muscle. *RXR*, retinoid X receptor; *ECR*, ecdysone receptor; *E75*, ecdysone-induced protein 75. Capital letters indicate significant differences in the different salinities under high-culture-density conditions ($p < 0.05$). Lowercase letters indicate significant differences in the different salinities under low-culture-density conditions ($p < 0.05$) (the same letters or no letters indicate that there are no significant differences, with $p > 0.05$). The asterisk (*) indicates significant differences between high-density and low-density treatments under the same salinity conditions ($p < 0.05$).

### 3.5. Energy-Related Genes

The expression of the ATP-$\alpha$ subunit gene under high salinity and stocking density stress is shown in Figure 5. The expression of the ATP-$\alpha$ gene showed a gradual decrease followed by a gradual increase as salinity increased. The expression of the ATP-$\alpha$ subunit gene was significantly higher under low-density stress than high-density stress in the 28 g/L salinity group; however, the situation was reversed in the 44 g/L salinity group ($p < 0.05$). The expression level of the ATP-$\alpha$ subunit gene was significantly lower at a salinity of

36 compared to the other two salinities, in both high and low-density farming conditions. This suggests that at a salinity of 36, the shrimp have the lowest energy expenditure.

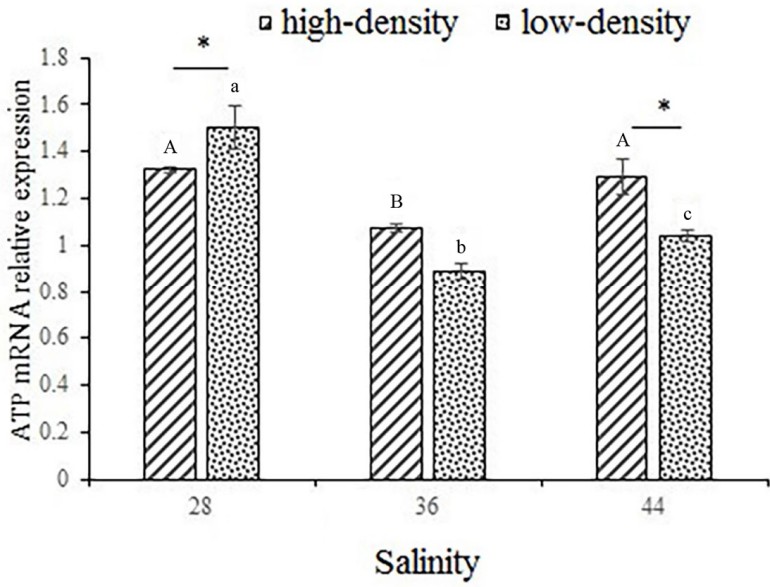

**Figure 5.** The expression of ATP-α subunit energy-related gene in muscle. Capital letters indicate significant differences in the different salinities under high-culture-density conditions ($p < 0.05$). Lowercase letters indicate significant differences in the different salinities under low-culture-density conditions ($p < 0.05$) (the same letters or no letters indicate that there are no significant differences, with $p > 0.05$). The asterisk (*) indicates significant differences between high-density and low-density treatments under the same salinity conditions ($p < 0.05$).

*3.6. Energy Balance*

As can be seen in Table 3, energy allocation was significantly influenced by salinity and culture density. With the increase in salinity, the ratio of energy for growth and the excretion of fecal energy for feed intake increased significantly in the high-density group; however, the higher the salinity, the lower the energy used for growth. The proportion of energy consumed by respiration and excretion gradually decreased as salinity increased, while the proportion of molting energy first decreased and then increased ($p < 0.05$). In the low-density group, the proportion of growth energy decreased significantly as salinity increased, and the proportions of respiratory and molting energy decreased significantly ($p < 0.05$), while there was no significant difference in excretory energy. In a comparison of high and low densities, it was noted that the 28 g/L salinity group's stocking densities had a significant effect on growth energy, respiratory energy, excretory energy, and molting energy ($p < 0.05$). In the 36 g/L salinity group, stocking densities only had significant effects on respiratory energy, while the other energy shares were not significantly different. In the 44 g/L salinity group, stocking densities had significant effects on the growth energy and respiratory energy of *L. vannamei* but no significant differences were found in other energy shares. This means that whether it is high-density or low-density farming, shrimp can obtain a more favorable energy supply for growth under the salinity of 36. This suggests that the salinity of 36 is more beneficial for the growth of shrimp, enabling them to utilize energy more effectively to support their growth.

**Table 3.** The effect of density on the energy budget of *Litopenaeus vannamei* under different salinities.

| Salinity | Density (shrimp/m³) | Feeding Energy | Growth Energy/Feeding Energy | Respiration/Feeding Energy | Fecal Energy/Feeding Energy | Excretion Energy/Feeding Energy | Molting Energy/Feeding Energy |
|---|---|---|---|---|---|---|---|
| 28 g/L | 600 | 20.79 ± 2.62 aA | 10.19 ± 1.2 aA | 71.54 ± 1.8 aA | 9.51 ± 0.8 a | 7.02 ± 0.4 aA | 1.74 ± 0.02 aA |
| | 300 | 30.07 ± 1.0 aB | 15.38 ± 0.52 aB | 64.75 ± 1.0 aB | 11.22 ± 0.7 a | 6.27 ± 0.5 B | 2.38 ± 0.08 aB |
| 36 g/L | 600 | 25.12 ± 1.2 bA | 11.29 ± 0.58 a | 69.75 ± 1.6 aA | 10.54 ± 0.8 a | 6.99 ± 0.1 a | 1.43 ± 0.07 b |
| | 300 | 43.16 ± 3.2 bB | 13.44 ± 2.7 b | 66.79 ± 2.0 aB | 11.38 ± 0.6 a | 6.69 ± 0.5 | 1.7 ± 0.03 b |
| 44 g/L | 600 | 18.52 ± 1.5 cA | 10.73 ± 0.7 b | 68.34 ± 0.6 bA | 13.2 ± 0.4 b | 5.75 ± 0.7 b | 1.99 ± 0.04 c |
| | 300 | 36.19 ± 3.0 abB | 10.04 ± 1.5 c | 66.22 ± 2.2 bB | 15.78 ± 0.8 b | 5.94 ± 0.6 | 2.02 ± 0.03 a |
| *p*-values | Salinity (S) | 0.219 | 0.037 | 0.013 | 0.436 | 0.048 | 0.035 |
| | Density (D) | 0.042 | 0.026 | 0.021 | 0.542 | 0.322 | 0.163 |
| | S × D | 0.224 | 0.049 | 0.029 | 0.683 | 0.416 | 0.217 |

The capital letters in the table above represent the significance of differences between treatments with different densities under the same salinity; the lowercase letters represent the significance of differences between treatments with different salinities under the same density.

## 4. Discussion

Shrimp, a high-quality source of protein, are experiencing an increasing demand worldwide. Determining how to enhance shrimp production is an urgent priority. Expanding farming areas and increasing stocking density are two highly important measures. The vast salt fields and high-salinity water areas in the world are a treasure trove awaiting development. High stocking density is employed to maximize production and optimize land and water resource utilization, and the Pacific white shrimp is an excellent breed known for its broad salt tolerance and high resistance to various stressors. Thus, we aimed to investigate the impact of two crucial factors, high salinity and stocking density, on *L. vannamei*.

### 4.1. Effects of High Salinity and Stocking Density Stress on Digestive Enzyme Activity

The ability of organisms to digest feed and absorb nutrients is mainly reflected by their digestive enzyme activity in vivo. Most of the previous research reports focused on the digestive capacity of *L. vannamei* under low-salt conditions [38]. The activities of trypsin, amylase, superoxide dismutase, and catalase in *L. vannamei* under a salinity of 3.0 g/L were higher than those in the salinity group of 17 g/L [3,39]. Previous research has shown that under low-salt conditions of 5 g/L, 10 g/L, and 15 g/L, the activities of amylase, lipase, and trypsin of *L. vannamei* decrease with increasing salinity [40]. This pattern still holds true in the high-salt testing conducted in this study. The present study obtained similar results: the activities of pepsin, amylase, and lipase in shrimp showed an overall weakening trend as salinity increased. It was also found that there was an overall effect among the three enzymes; i.e., the trend of the three enzyme activities showed consistency, and this trend was more significant in the high-density groups. However, under the high-density-culture and high-salinity conditions, the activities of the three enzymes were higher in the high-density group than those in the low-density group. This observation contradicts the trend observed in previous studies on *Palaemonetes sinensis* [3]. It is speculated that the possible reason for this is that high salinity breaks the balance of the original density stress, resulting in the high-density population being more inclined to seek food to maintain their own metabolism under high-salinity conditions.

### 4.2. Effects of High Salinity and Culture Density Stress on Growth

It is generally believed that when seawater salinity is in the optimal salinity range of shrimp or close to the isosmotic point, it is most conducive to the growth of shrimp [41,42]. Shrimp require more energy for the regulation of osmotic pressure balance in their bodies, and the energy that would have been used for growth is consumed, resulting in the inhibition of the growth, digestion, and immunity of these organisms [43–45]. Culture density is another influencing factor that has been reported both at home and abroad. The growth of shrimp will be inhibited when the stocking density is too high, which can lead to

a lower weight gain rate and a reduced survival rate [46]. In this study, the low-density-cultures of *L. vannamei* were superior to the high-density cultures in terms of body weight and specific growth rate, a finding similar to the results of previous studies. In both farming densities, the growth performance of 36 g/L salinity shrimps was superior to the other two salinities. Previous studies have shown that the growth of *L. vannamei* is best at salinities around 33–40 g/L [46,47]. The results of this study are similar to those of the previous study. Therefore, the optimal farming salinity for *L. vannamei* would be within a range centered around 36 g/L.

Ribosomal proteins are involved in various processes and play important roles in the growth of organisms [48]. SNRPG (Small nuclear ribonucleoprotein polypeptide G) is mainly involved in the processing of mRNA during the translation phase of proteins and promotes protein synthesis [49]. RPL7 (ribosomal protein L7) is the large subunit protein of ribosomal organelles, and RPL7 is mainly involved in gonadal development in mammals and fish [50]. L18 (Ribosomal protein L18) is a component of the large ribosomal subunit, and its expression is relatively stable in different growth stages or different tissues of *Anastrepha obliqua* and *Portunus trituberculatus* [51,52]. In this study, three genes were selected for an analysis of the growth traits of *L. vannamei* under high salinity and stocking density stress. It was found that the expression of *SNRPG*, *RPL7*, and *L18* was generally higher in the low-density treatments than in the high-density treatments. The potential reason could be that high stocking density induces chronic stress, triggering physiological and oxidative stress, resulting in increased energy consumption and growth inhibition. Similar findings have also been observed in studies on fish [53].

Molting is crucial for the growth of shrimp. Investigating the physiological processes involved in molting can provide insights into the growth statuses of shrimp. The synthesis of ecdysteroids during molting is subject to the combined action of several hormones, occurring through a series of processes such as the transmission of transcription factors such as RXR (retinoid X receptor), ECR (ecdysone receptor), and E75 to finally complete molting [54]. In our study, it was observed that in the low-density group, the expression levels of three molting-related genes decreased with increasing salinity. This suggested that the increase in salinity inhibited the expression of molting-related genes, which was detrimental to shrimp molting and growth. Similar patterns have also been observed in studies on juvenile shrimp, indicating that changes in salinity have a significant impact on their molting and growth [24]. However, a new pattern emerged in the high-density treatment, revealing that the expression levels of all three molting-related genes in the salinity 36 g/L experimental group were higher than the other two salinity groups. This suggested that selecting an appropriate salinity level was beneficial for the molting and growth of shrimp under high-density farming conditions. In the exploration of inland ground saline water aquaculture, it has been found that selecting an appropriate salinity level under specific stocking density conditions is beneficial for shrimp growth [55].

### 4.3. Effects of High Salinity and Culture Density Stress on Energy Balance

Crustaceans have a good ability to regulate osmotic pressure [56]. This regulation exhibits duality. When the salinity is higher than that of crustaceans in vivo, excess salt needs to be secreted out of the body in order to keep the water in the body from being lost; on the contrary, the excess water needs to be excreted from the body. The processes of salt secretion and water loss require a great deal of energy consumption. Studies have shown that the respiratory metabolism of shrimp and crabs consumes the least amount of energy at the isotonic point [42]. The effect of density on the energy of aquatic organisms under the conditions of artificial culturing is mainly due to the increased movement and struggling of the shrimp in an attempt to obtain food and coordinate interspecific struggles, resulting in an increased respiratory metabolic capacity and a reduction in the energy used for growth [57]. In this study, the results showed that the high-density-culture group used significantly more energy for respiration than the low-density-culture group, similar to the conclusions obtained in the above-mentioned study. This meant that excessively high

stocking density resulted in unnecessary energy consumption, which was detrimental to the growth of shrimp. This conclusion has also been validated in studies on tiger shrimp *Penaeus monodon* [58]. Additionally, our research found that the order of energy types in the experiment as a percentage of the feeding energy was as follows: respiration consumption > growth accumulation > energy contained in feces > excretion consumption energy > molting consumption energy. This study provides valuable insights and references for future research on the impact of stocking density and salinity stress on shrimp energy balance, as well as the formulation of shrimp farming strategies.

## 5. Conclusions

Our research confirmed that the synergistic effect of salinity and stocking density on *L. vannamei* was significant. The synergistic effect of the two factors resulted in different changes in the growth performance, digestive capacity, and energy metabolism of shrimp. We recommend selecting a farming salinity within the range of 36 and its vicinity and avoiding excessively high breeding density (greater than 600) because excessively high salinity (44 g/L) could weaken the digestive capacity and affect the energy balance of shrimp, and an excessively high stocking density (600 shrimp/m$^3$) could lead to a reduced growth rate of shrimp. The scientific and reasonable control of salinity and stocking density can effectively improve the efficiency of shrimp aquaculture. This study provides valuable experimental data for the development of the intensive culturing of *L. vannamei* under high-salinity stress.

**Author Contributions:** F.L.: conceptualization, formal analysis, writing—original draft, project administration, funding acquisition, writing—review and editing. J.S.: methodology, formal analysis, and visualization. J.L.: formal analysis and writing—original draft. L.S.: resources and data curation. C.L. and X.W.: investigation. L.Z. and P.H.: resources. Y.C. and R.W.: investigation and resources. Z.W.: funding acquisition, investigation, and methodology. Y.L.: conceptualization, supervision, project administration, funding acquisition and writing—review and editing. All authors have read and agreed to the published version of the manuscript.

**Funding:** The work was supported by the National Key Research and Development Program of China (2022YFD2400203); the National Science Foundation of China (31902407, 32102762); the earmarked fund for the Modern Agro-industry Technology Research System in Shandong Province (SDAIT-13-03); the Key R&D Program of Shandong Province (2021LZGC027); the High-level Talents Research Fund of Qingdao Agricultural University (665/1122015); and the "First Class Fishery Discipline" program in Shandong Province.

**Institutional Review Board Statement:** All treatments in this study were undertaken strictly in accordance with the guidelines of the Animal Experiment Ethics Committee of Qingdao Agriculture University, which also approved the protocol in May 2020 (Approval Code: 2020-026).

**Data Availability Statement:** Data are contained within the article.

**Acknowledgments:** We would like to express our gratitude to Changyi Haijingzhou Biotechnology Co., Ltd. (Weifang, China) for providing the experimental animals for this study.

**Conflicts of Interest:** The authors declare no conflicts of interest.

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
