# Peer review of "Assessing the Interactive Effects of High Salinity and Stocking Density on the Growth and Stress Physiology of the Pacific White Shrimp Litopenaeus vannamei"

_fishes, doi:10.3390/fishes9020062_

Round 1
Reviewer 1 Report
Comments and Suggestions for Authors
The manuscript titled "Assessing the interactive effects of high salinity and stocking density on the growth and stress physiology of the Pacific white shrimp Litopenaeus vannamei" investigates the combined effects of high salinity and stocking density on Litopenaeus vannamei in terms of growth performance, digestibility, and energy balance. In general, the manuscript suffers from many defects, such as statistical analysis, writing, and interpretation of the data as well as a lack of focus.
Major Concerns:
1. The language is often cumbersome or imprecise. The manuscript contains many grammatical, typographic, and styling errors. Revision of the manuscript by a native English speaker is highly recommended.
2. In Section 2.8, it is stated that the results were analyzed using a one-way ANOVA. However, throughout the text, it is clear that a two-way ANOVA was actually conducted. I agree that the experimental design is factorial, and therefore, the statistical analysis section needs to be revised to ensure consistency with the analysis conducted in the manuscript.
3. In general, the manuscript discusses the effects of salinity and stocking density separately, without adequately addressing their interaction. Although the interaction effect was identified as significant, the manuscript fails to mention the specific optimum salinity level for each density. It is crucial to include this information in the results section, thoroughly discuss it, and provide a final recommendation based on the interaction between salinity and stocking density, rather than considering each factor independently.
4. The presentation of the data in the current manuscript is insufficient to provide a comprehensive understanding of the results. For instance, Table 2 only presents the significance only, while the results are not provided. Additionally, in Figure 1, only two effects, salinity and density, are shown, neglecting the interaction effect. I recommend that the data be presented in tables, similar to Table 3, to ensure all relevant results are included. However, if the results are to be presented in figures, it is imperative that the interaction effect is also incorporated.
5. In all tables presented, it is recommended to provide the exact p-values instead of using "NS" or "*". Including the precise p-values will enhance the clarity and transparency of the statistical analysis conducted in the study.
Additional comments:
1. Clarify the objective and hypothesis of the study at the end of the Introduction section.
2. Line 90; Add more descriptions about experimental tanks.
3. Table 1; Add gene accession number or reference for each primer. Also, describe all abbreviations (genes) used in the table footnote.
4. The instruments (e.g. in lines 103, 104, … etc) and chemicals (e.g. in lines 130-134, … etc) must contain all full information such as model, company, city, country.
5. Line 110; Were the shrimp anesthetized before sampling?
6. Line 148Add reference to this method.
7. why was the data on feed intake and feed conversion not provided?
8. Discussion only rely on speculation and not rely on solid science. All estimated parameters should be discussed in more detail with more depth. Also, the discussion must be fortified with possible attributions supported by references.
Comments on the Quality of English Language--
Author Response
Dear Reviewers,
Thank you for your hard work and valuable suggestions on this manuscript. We have carefully considered your comments and made revisions accordingly. Please find the detailed responses and revised manuscript in the re-submitted files.
Kind regards,

Reviewer 2 Report
Comments and Suggestions for Authors
You can find my comments and questions in the attached MS. Please consider my critics for making an article of higher quality.

No comments.
Author Response

(The authors gave the same response as above.)

Round 2
Reviewer 1 Report
Comments and Suggestions for Authors
The authors have made extensive revisions to the manuscript, addressing several concerns raised during the previous review. The manuscript has significantly improved in terms of clarity and presentation. However, there are still a few remaining comments that need to be addressed before the manuscript can be considered suitable for publication. The following points outline these concerns:
1) The exact p-value should be provided for each factor (salinity, stocking density, and interaction) in all tables and figures. While the revised manuscript mentions the inclusion of p-values, it is crucial to provide the specific values rather than stating their existence without further elaboration.
2) In Table 2, it is recommended to present the p-values as three decimals rather than using E notation. For example, instead of displaying "4.39E-01," it is preferable to present it as "0.439."
Author Response
Dear Reviewers,
Thank you again for your hard work and valuable suggestions on our manuscript. We have carefully considered your further comments and made revisions accordingly. Please find the detailed responses and revised manuscript in the re-submitted files.
Kind regards,

Reviewer 2 Report
Comments and Suggestions for Authors
The MS improved significantly and can be published.
Author Response
Dear Reviewers,
Thank you once again for your hard work on this manuscript.
Kind regards,
